# NOVA-GS: Noise-Aware View-Consistent Gaussian Splatting for Low-Light Novel View Synthesis

Shaurya Pavan A     Vemunuri Divya Madhuri     Yash Pradeep Gawande     Kaushik Mitra

Indian Institute of Technology Madras

{mm24b048, ee24d004, me21b062}@smail.iitm.ac.in, kmitra@ee.iitm.ac.in

## Abstract

*Reconstructing 3D scenes under real-world low-light conditions remains challenging due to severe sensor noise, low signal-to-noise ratios, and degraded photometric consistency, which destabilize geometry estimation and novel view synthesis. Existing approaches often rely on well-lit reference data for reliable Structure-from-Motion (SfM) initialization under degraded inputs or apply per-view enhancement methods that introduce cross-view inconsistencies. To address these limitations, we propose **NOVA-GS**, a unified noise-aware framework for low-light 3D Gaussian Splatting that subsumes enhancement, denoising, and geometry optimization within a single process. Our method leverages VGGT-based feed-forward estimation to obtain robust camera poses and geometry directly from degraded inputs, eliminating the need for SfM. Building on this initialization, NOVA-GS integrates three coupled components: a structure-aware enhancement module for exposure correction, a self-supervised denoising module with blind-spot masking for pseudo-supervision, and a consistency-driven Gaussian Splatting optimization enforcing cross-view geometric coherence. We further introduce a noise-guided spherical harmonic regularization to suppress view-dependent artifacts in noisy regions. Extensive experiments on diverse real-world low-light datasets demonstrate improved geometric fidelity, color consistency, and robustness without requiring paired supervision or well-lit references.*

## 1. Introduction

Reliable 3D reconstruction and novel view synthesis are indispensable for autonomous systems in environments where controlled illumination cannot be guaranteed, including nighttime autonomous driving, low-light robotic perception, nocturnal infrastructure monitoring, underground mining, and search-and-rescue operations. While variants of Neural Radiance Fields (NeRF) [1, 2, 18, 20, 26] and 3D Gaussian Splatting (3DGS) [10, 11, 15] have emerged as dominant paradigms, both struggle in these settings. They fundamentally assume well-lit inputs where pixel intensities provide reliable cues of scene radiance and geometry, an assumption that breaks down in challenging real-world low-light conditions.

Several methods attempt to address adverse illumination within NeRF and Gaussian splatting based frameworks. NeRF-based approaches introduce radiometric modeling or illumination decomposition [4, 14, 19, 22, 27], but remain computationally intensive and unsuitable for real-time rendering. Recent low-light Gaussian Splatting variants improve robustness through illumination priors or decomposition strategies [5, 9, 25, 28, 35, 36]. However, many approaches assume reliable Structure-from-Motion (SfM) initialization [24], typically obtained from well-lit images. In low-light conditions, SfM from degraded inputs becomes unreliable, with methods such as COLMAP often failing or leading to poor initialization and unstable optimization.

A common alternative is to enhance images prior to reconstruction; however, per-view enhancement methods [3, 6, 7, 16, 17, 33] ignore cross-view consistency and introduce photometric inconsistencies that lead to geometric artifacts. Conversely, reconstruction methods operating directly on degraded inputs often fail to disentangle noise from true structure, especially under severe noise [5]. These limitations motivate a unified optimization framework that simultaneously addresses enhancement, denoising, and geometry.

To this end, we propose **NOVA-GS**, a unified, unsupervised low-light 3D Gaussian Splatting framework that subsumes image refinement, noise-aware denoising, and 3D Gaussian Splatting into a single optimization process. By enforcing cross-view consistency and suppressing noise propagation during rendering, our method reduces artifacts and improves structural coherence, enabling robust novel view synthesis directly from extreme low-light inputs without requiring paired supervision or preprocessing. Built on the efficient 3DGS paradigm [11], it achieves real-time ren-

dering, faster training than NeRF-based and most of the Gaussian Splatting methods, and demonstrates strong performance across diverse indoor and outdoor datasets under moderate to extreme low-light conditions (see Section A of the supplementary material). Our main contributions are summarized as follows:

- **Robust Initialization without SfM.** Given the failure of traditional SfM pipelines in low-light and noisy conditions, we leverage VGGT for geometry initialization to achieve stable reconstruction.
- **Noise-Aware Self-Supervised Denoising for 3D Reconstruction.** We introduce a novel targeted blind-spot denoising strategy that leverages spatial noise priors derived from high-frequency components, coupled with a Deep Attention-ResUNet architecture to generate clean pseudo-supervision while preserving structural details critical for 3D optimization.
- **Noise-Guided Spherical Harmonic Regularization.** We introduce a novel regularization scheme that maps 2D noise estimates to 3D Gaussians, selectively penalizing higher-order spherical harmonic components in noisy regions to prevent view-dependent artifacts.

## 2. Related Work

### 2.1. Low-Light Novel View Synthesis with NeRF

NeRF-based methods have been extended to low-light settings through explicit illumination modeling. LLNeRF [27] decomposes radiance into illumination-dependent and view-independent components, but often produces unnatural colors. Aleth-NeRF [4] models light attenuation via a concealing field, assuming consistent attenuation across scenes. $I^2$-NeRF [14] incorporates physically grounded media interactions within an unsupervised framework, but leverages ground-truth illumination statistics, introducing implicit supervision that limits generalization.

However, these methods rely on volumetric MLP-based rendering [1, 2, 18, 20], resulting in slow training and non-real-time inference, and depend on accurate SfM poses [24], which are unreliable under low-light conditions.

### 2.2. Low-Light Gaussian Splatting Methods

Recent works extend 3D Gaussian Splatting to low-light settings with explicit illumination modeling. LLGS [28] introduces a decomposable color representation for joint enhancement and reconstruction. LL-Gaussian [25] models intrinsic reflectance and illumination with a dedicated initialization strategy, also uses diffusion-priors for supervision. Luminance-GS [5] captures luminance variations via per-view color mapping and curve estimation, but does not explicitly handle noise under extreme low-light. LITA-GS [36] incorporates illumination-invariant priors with progressive denoising for reference-free optimization.

Despite these advances, many methods rely on stable initialization from well-lit inputs and lack robustness under extreme low-light.

### 2.3. 2D Low-Light Enhancement Methods

Low-light image enhancement methods improve visibility via learned illumination adjustment and non-linear tone mapping [7, 16, 17, 33]. However, they operate independently per view and lack multi-view photometric consistency. When used for 3D reconstruction, these inconsistencies accumulate and degrade geometric optimization, highlighting the need for multi-view geometric consistency constraints.

**SfM Initialization in Low-Light Reconstruction.** Accurate camera pose and geometry initialization remain challenging in low-light conditions due to degraded feature quality and unstable matching. Traditional SfM pipelines such as COLMAP [24], GLOMAP [21], and hierarchical localization frameworks like hloc [23] rely on robust feature detection and correspondence matching, which often degrade under low illumination and noise. Recent learning-based approaches, including DUSt3R [31], Mast3R [13], VGGT [30], and VGGSfM [29], adopt feed-forward inference to improve robustness under such challenging conditions. For instance, LLGaussian [25] employs DUSt3R for initialization; however, such approaches typically require additional post-processing to refine and prune the resulting point cloud. In practical low-light settings, where only degraded inputs are available, initialization remains noisy and uncertain, often propagating errors into downstream 3D optimization. To address these challenges, we leverage VGGT-based initialization for improved robustness.

## 3. Preliminaries

### 3.1. 3D Gaussian Splatting as MCMC

**3D Gaussian Splatting (3DGS).** 3D Gaussian Splatting (3DGS) represents a 3D scene using a collection of anisotropic Gaussians parameterized by mean $\mu_i$, covariance $\Sigma_i$, opacity $\alpha_i$, and view-dependent color $c_i$. During rendering, these Gaussians are projected onto the image plane and composited using $\alpha$-blending.

The blended color at pixel $\mathbf{x}$ is computed as:

$$C(\mathbf{x}) = \sum_{i=1}^{N} c_i \, \alpha_i(\mathbf{x}) \prod_{j=1}^{i-1} (1 - \alpha_j(\mathbf{x})), \qquad (1)$$

where $\alpha_i(\mathbf{x})$ is evaluated from $(\mu_i, \Sigma_i)$.

**3DGS as Markov Chain Monte Carlo (MCMC).** To avoid heuristic densification, 3DGS-MCMC [11] reformu-

lates Gaussian optimization as an MCMC sampling process using Stochastic Gradient Langevin Dynamics (SGLD).

## 3.2. VGGT: Visual Geometry Grounded Transformer

In extremely low-light images, traditional SfM methods (e.g., COLMAP [24]) fail to extract reliable correspondences. To address this, we leverage VGGT [30] for direct geometry initialization.

Given $N$ unposed low-light images $\{I_{\text{dark},i}\}_{i=1}^{N}$, VGGT predicts camera parameters $g_i$, depth maps $D_i$, tracking features $T_i$, and dense 3D point maps $P_i$:

$$\{(g_i, D_i, P_i, T_i)\}_{i=1}^{N} = F_{\text{VGGT}}\left(\{I_{\text{dark},i}\}_{i=1}^{N}\right). \quad (2)$$

The predicted dense point maps and camera parameters serve as robust initialization for 3D Gaussians, removing the need for classical SfM.

## 4. Methodology

### 4.1. Overview

As illustrated in Fig. 1, we propose an unsupervised, unified optimization framework for 3DGS under extreme low-light conditions. Given $N$ low-light images $\{I_{\text{dark},i}\}_{i=1}^{N}$ alongside initial point clouds and camera poses obtained from VGGT [30], our pipeline processes the input views in three concurrent stages. First, a structure aware enhancement module brightens the input to a target exposure using a Differentiable Guided Filter (DGF). Second, an unsupervised targeted blind-spot network filters each noisy enhanced image independently, synthesizing a clean pseudo-ground-truth target to supervise the 3DGS geometry. Finally, the 3D Gaussians are optimized against this clean target, strictly constrained by cross-view geometric consistency and noise-guided spherical harmonic losses to prevent multi-view artifacts.

### 4.2. Enhancement Module

Direct enhancement of extreme low-light images amplifies high-frequency noise, leading to over-exposed artifacts and unstable geometry. Applying gamma correction or intensity scaling directly to original pixel values increases both illumination and noise, making exposure correction unreliable. To address this issue, we perform per-view enhancement on a structurally smoothed representation of each input image, denoted as $I_{\text{dark}}$, rather than directly modifying noisy pixel intensities.

We first compute a low-frequency image $L$ from $I_{\text{dark}}$ using a Differentiable Guided Filter [32]. The guided filter performs edge-aware smoothing, suppressing high-frequency noise while preserving structural boundaries. The resulting image $L$ acts as a smooth **brightness map** that

captures the overall illumination distribution of the scene. By reducing local intensity fluctuations caused by noise and texture, exposure correction is guided by stable structural brightness cues instead of noisy pixel-level variations. Additional details are provided in Section B.1 of the supplementary material.

The brightness map ($L$) is then enhanced using a gamma transformation:

$$L_{enh} = L^{\gamma} \quad (3)$$

where $\gamma$ is optimized to push the mean brightness level toward a target exposure value $\tau = 0.5$. The optimal $\gamma$ is computed using a binary search that minimizes the difference between the mean brightness of $L_{enh}$ and the target exposure level. Enhancing the smoothed brightness map ensures that exposure correction operates purely on the low-frequency scene structure, preventing the uncontrolled amplification of high-frequency noise.

To propagate this exposure correction back to the original image, we compute a spatial brightness ratio map:

$$R = \frac{L_{enh}}{L + \eta} \quad (4)$$

where $\eta$ is a small constant for numerical stability. This ratio represents the multiplicative exposure adjustment required at each spatial location.

The final enhanced image is obtained as:

$$I_{enh} = \min(I_{\text{dark}} \odot R, 1) \quad (5)$$

where $\odot$ denotes element-wise multiplication. This ratio-based formulation performs exposure correction using low-frequency brightness structure, preserving textures and geometric cues while avoiding direct modification of noisy pixel intensities (see Fig. 5 and Section B.2 in the supplementary material). In contrast to direct gamma enhancement, this approach produces spatially consistent brightness adjustments.

### 4.3. Denoising Pipeline

Although the initial per-view enhancement stabilizes global exposure, the resulting image $I_{enh}$ inherently retains significant signal-dependent camera noise. Optimizing 3D Gaussian Splatting directly on this noisy supervision causes the geometry and spherical harmonic coefficients to overfit to noise patterns, manifesting as geometric floaters and severe multi-view inconsistencies. To prevent this degradation, we introduce a self-supervised denoising pipeline that synthesizes a clean pseudo-ground-truth target for each view prior to 3DGS optimization. While built upon the theoretical foundation of Noise2Void (N2V) [12], our framework introduces two critical extensions to handle the severe, spatially varying noise typical in low-light conditions: a **Targeted Blind-Spot Masking** strategy driven by high-frequency spatial priors, and a **Deep Attention-ResUNet**

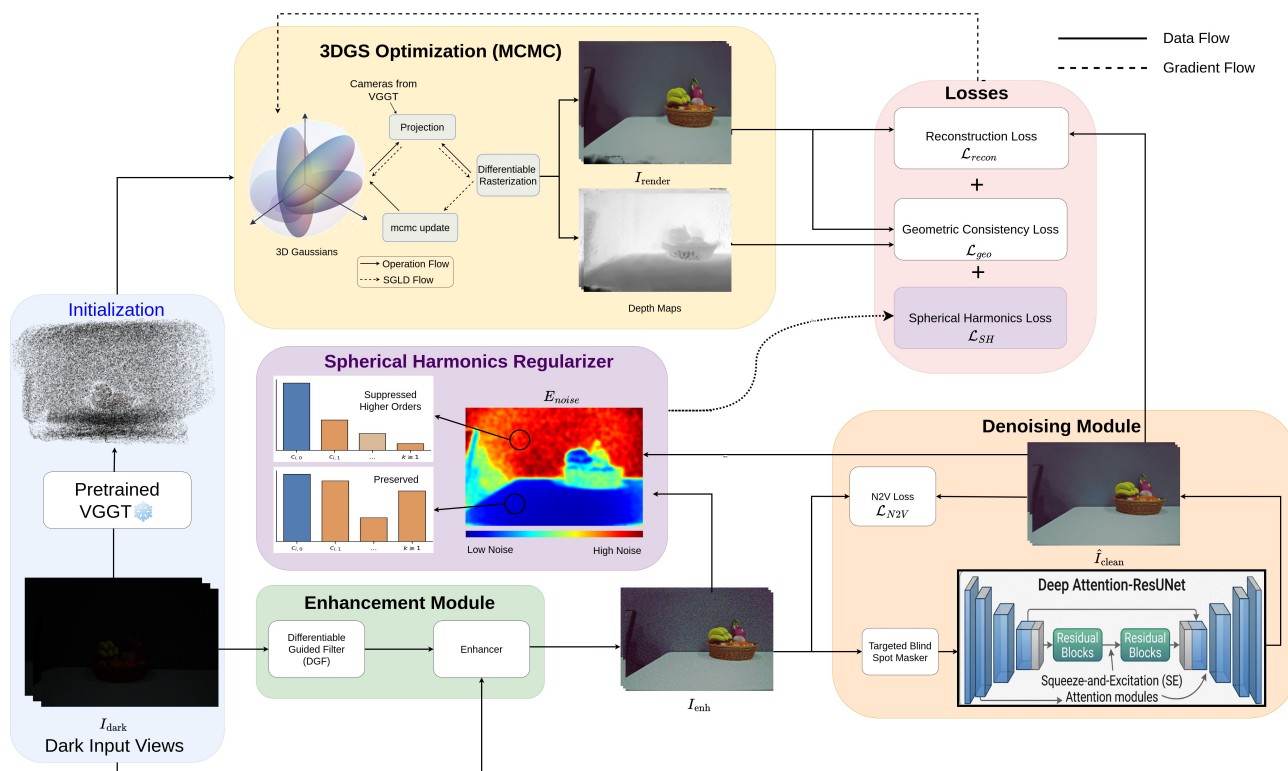

Figure 1. **Overview of our proposed unified optimization framework (NOVA-GS).** Given low-light inputs, our pipeline concurrently performs preliminary denoising and enhances in enhancement module, self-supervised denoising using a targeted blind-spot network, and consistency-aware 3D Gaussian Splatting optimization constrained by geometric, and noise-guided spherical harmonic losses.

architecture designed to dynamically filter noise while preserving delicate structural geometries.

### 4.3.1. Targeted Blind-Spot Masking

Standard N2V employs uniform random masking, which may over-sample smooth regions while insufficiently focusing on structurally complex and noise-prone areas. To address this limitation, we introduce a targeted blind-spot masking strategy that prioritizes high-frequency regions during self-supervised training.

We first extract a high-frequency spatial prior by applying a Laplacian operator to the grayscale representation of the enhanced image $I_{\text{gray}}$:

$$I_{\text{Lap}} = \nabla^2 I_{\text{gray}} \tag{6}$$

The magnitude of this response is normalized across spatial dimensions to form a probability distribution:

$$P(u, v) = \frac{|I_{\text{Lap}}(u, v)|}{\sum_{u,v} |I_{\text{Lap}}(u, v)|} \tag{7}$$

Blind-spot indices are then sampled from a mixture of this distribution and uniform random sampling, producing

a binary mask $M(p)$ over pixel coordinates $p$. This encourages the network to focus denoising on structurally complex and noisy regions while maintaining stable optimization in smoother areas.

### 4.3.2. Deep Attention-ResUNet.

To recover fine details from these heavily masked regions, we replace the baseline U-Net with a Deep Attention-ResUNet (for architecture details refer to Section B.3 in the supplementary material). This architecture features a four-level encoder-decoder equipped with Residual Blocks to prevent the loss of high-frequency boundaries during downsampling. Furthermore, we integrate Squeeze-and-Excitation (SE) attention modules [8] into the residual blocks. These modules dynamically recalibrate channel-wise feature responses, actively suppressing noise-dominant channels while exciting those containing critical geometric structures.

### 4.3.3. Self-Supervised Denoising Loss

To train the network without paired clean data, we employ a masked Mean Squared Error (MSE) loss evaluated only at the selected blind spots. Let $M(\mathbf{p}) = 1$ if a pixel $\mathbf{p}$ corresponds to a masked blind spot, and $0$ otherwise. The de-

noising loss $\mathcal{L}_{N2V}$ measures the error between the network prediction $\hat{I}_{clean}$ and the original noisy input $I_{enh}$:

$$\mathcal{L}_{N2V} = \frac{\sum_{\mathbf{p}} M(\mathbf{p}) \|\hat{I}_{clean}(\mathbf{p}) - I_{enh}(\mathbf{p})\|_2^2}{\sum_{\mathbf{p}} M(\mathbf{p})} \quad (8)$$

Since the network is explicitly blinded to the true center pixel value $I_{enh}(\mathbf{p})$, it must infer the underlying structure from the surrounding spatial context. This encourages the model to recover meaningful structural information while suppressing spatially independent noise. The masked loss serves as the sole optimization objective for the denoiser.

### 4.4. Reconstruction Loss

To optimize the 3D Gaussian geometry and view-dependent appearance without clean reference images, we supervise the rendered image $\hat{I}_{render}$ using the pseudo-clean target $\hat{I}_{clean}$ produced by the denoising module. We compute an $L_1$ reconstruction loss between $\hat{I}_{render}$ and $\hat{I}_{clean}$ The final reconstruction loss combines $L_1$ with a structural similarity term:

$$\mathcal{L}_{recon} = (1 - \lambda)\mathcal{L}_1 + \lambda(1 - \text{SSIM}(\hat{I}_{render}, \hat{I}_{clean})) \quad (9)$$

### 4.5. Consistency-Aware Optimization

Since per-view enhancement and denoising operate independently, they may introduce structural inconsistencies across views. To enforce global scene consistency, we constrain the 3DGS optimization using a geometric consistency loss.

#### 4.5.1. Depth-Guided Reprojection

To enforce multi-view photometric consistency, we warp a source view $s$ into a target view $t$ using the rendered 3DGS depth maps and known camera poses. This depth-guided reprojection produces the warped source image $\hat{I}_s$ in the target coordinate frame.

To restrict comparisons to reliable overlapping regions, we filter out out-of-bounds and negative-depth pixels. Occlusions are further resolved by comparing the projected source depth $z_t$ with the rendered target depth $d_t(\mathbf{p}_t)$ at each pixel. A pixel is considered valid and non-occluded if:

$$\frac{|d_t(\mathbf{p}_t) - z_t|}{z_t} < 0.1 \quad (10)$$

These operations produce a binary mask of reliable pixels, denoted as $M_{\text{geo}}(\mathbf{p})$.

#### 4.5.2. Confidence-Aware Photometric Loss

Under severe low-light conditions, not all reprojected pixels are equally reliable. To prevent poorly aligned regions from corrupting optimization, we predict a per-pixel confidence map $C$ using a lightweight convolutional network.

The network takes the warped source image and target image as input to estimate reprojection reliability. Additional implementation details are provided in Section B.4 of the supplementary material.

$$C = \sigma(f_{\text{conf}}(\hat{I}_s, I_t)) \quad (11)$$

The photometric consistency loss over the valid pixel set $\Omega$ is weighted using the predicted confidence:

$$\mathcal{L}_{\text{photo}} = \frac{1}{|\Omega|} \sum_{\mathbf{p} \in \Omega} C(\mathbf{p}) \left| I_t(\mathbf{p}) - \hat{I}_s(\mathbf{p}) \right| M_{\text{geo}}(\mathbf{p}) \quad (12)$$

To prevent the confidence map from collapsing to zero, we apply a negative log regularization term:

$$\mathcal{L}_{\text{conf}} = \lambda_c \frac{1}{|\Omega|} \sum_{\mathbf{p} \in \Omega} -\log(C(\mathbf{p})) M_{\text{geo}}(\mathbf{p}) \quad (13)$$

The final geometric consistency loss is defined as:

$$\mathcal{L}_{\text{geo}} = \mathcal{L}_{\text{photo}} + \mathcal{L}_{\text{conf}} \quad (14)$$

### 4.6. Noise-Guided Spherical Harmonic Regularization

Spherical Harmonics (SH) are utilized in 3DGS to model view-dependent appearance. However, residual 2D noise often manifests as rapid view-dependent color variations. The 3DGS framework inherently interprets this noise as a physical specular component, utilizing higher-order SH coefficients to fit it. This causes severe geometric artifacts known as floaters.

We mitigate this by applying a noise-adaptive SH penalty. We calculate a 2D spatial noise magnitude map $E_{noise}(\mathbf{p})$ by taking the channel-wise mean absolute difference between the noisy enhanced image and the clean pseudo-ground-truth target:

$$E_{noise}(\mathbf{p}) = \frac{1}{3} \sum_{c \in \{r,g,b\}} \left| I_{enh}^{(c)}(\mathbf{p}) - \hat{I}_{clean}^{(c)}(\mathbf{p}) \right| \quad (15)$$

For each visible 3D Gaussian $i$, we project its 3D center to the 2D image plane coordinate $\mu_i'$. We sample the noise map at these coordinates to assign a scalar noise weight $w_i = E_{noise}(\mu_i')$.

**Weighted SH Penalty.** The SH coefficients for a single Gaussian $i$ are represented as a set $C_i = \{c_{i,0}, c_{i,1}, \ldots, c_{i,K}\}$, where $c_{i,0}$ is the base color component representing the view-independent diffuse color, and components $c_{i,k}$ for $k > 0$ represent the higher-order, view-dependent emission.

To suppress the overfitting of noise without darkening the overall scene, we isolate and penalize only the higher-order SH coefficients. The final regularization loss is computed as the mean weighted $L_2$ norm (squared) of the high-order SH coefficients across all $N_{vis}$ visible Gaussians:

$$\mathcal{L}_{SH} = \frac{1}{N_{vis}} \sum_{i=1}^{N_{vis}} w_i \sum_{k=1}^{K} \|c_{i,k}\|_2^2 \qquad (16)$$

By directly tying the SH penalty to the noise magnitude, $\mathcal{L}_{SH}$ aggressively penalizes view-dependent variations in noisy regions (forcing those Gaussians to rely on their diffuse base color) while permitting higher-order SH optimization in the structurally clean, confident regions of the scene. Specular highlights remain consistent across views, while noise varies randomly between views, enabling the noise-weighted SH penalty to suppress noise-driven coefficients without affecting genuine view-dependent reflectance.

### 4.7. Final Optimization Objective

The complete training loss is defined as:

$$\mathcal{L}_{total} = \mathcal{L}_{recon} + \mathcal{L}_{geo} + \mathcal{L}_{SH} \qquad (17)$$

where $\mathcal{L}_{recon}$ is the reconstruction loss, $\mathcal{L}_{geo}$ is the geometric consistency loss, and $\mathcal{L}_{SH}$ is the noise-guided spherical harmonic regularization. All modules are jointly optimized in an end-to-end manner using gradient-based optimization.

## 5. Experiments

### 5.1. Datasets

We evaluate our method on four diverse low-light datasets. Refer supplementary section A for histogram plots. MVTV [34] spans a wide 16-bit range which consists of wide dynamic range, LLRS [25] is concentrated near zero with extreme noise, LOM [4] reflects low-light conditions, and LLNeRF [27] exhibits moderately low-light behavior with additional noise. These distributions highlight progressively challenging illumination regimes.

### 5.2. Baselines

We compare against recent 3D methods, including Aleth-NeRF [4], LLNeRF [27], I2-NeRF [14], Luminance-GS [5], Lita-GS [36], and hybrid 2D+3D pipelines (MBLLEN [16], URetinex-Net [33], SCI++ [17], Zero-DCE [7] + GS). All 3D methods rely on COLMAP[24] for initial sparse point cloud reconstruction and pose estimation; however, Aleth-NeRF, Luminance-GS, and I$^2$-NeRF additionally use ground-truth images to align luminance and contrast statistics.

For 2D + 3D baselines, we enhance images (MBLLEN, URetinex-Net, SCI++, Zero-DCE), estimate poses and initial pointcloud using VGGT [30], and train 3DGS-MCMC [11] for 7k iterations. We use a unified split with `llffhold = 8` across all methods for consistent novel view evaluation. SCI++, Zero-DCE results are shown in Appendix.

### 5.3. Implementation Details

Our framework is implemented using PyTorch and built upon a 3D Gaussian Splatting (3DGS MCMC [11]) pipeline. We initialize camera parameters and geometry using VGGT [30] predictions.

For optimization, we use Adam with a learning rate of $1e^{-3}$. The loss function consists of photometric consistency, confidence regularization, illumination consistency, and noise-guided spherical harmonic regularization. Training is performed for 7k iterations on an NVIDIA RTX 3090 GPU.

### 5.4. Results

We present visual comparisons in Fig. 2 across LOM (Sofa, Shrub), LLNeRF (Building, Staircase), and LLRS (Still2, Still3) scenes. Existing methods exhibit clear limitations: **Luminance-GS** produces artifacts such as floaters, ghosting, and over-smoothing (e.g., Building, Staircase), **Aleth-NeRF** shows competitive performance on LOM but suffers from structural degradation, including edge bleeding and incomplete geometry in LLNeRF (Still2, Still3), as well as color artifacts in Building and Staircase, and **Lita-GS** improves visibility but exhibits inconsistent color reproduction and local artifacts. 2D enhancement + GS pipelines (MBLLEN, URetinex-Net) further blur details and lack geometric consistency. In contrast, our method preserves fine structures (e.g., Building, Still2, Still3) while maintaining consistent and realistic colors, closely matching ground truth (e.g., Shrub, Sofa).

Quantitatively, as shown in Table 1, our method achieves the best performance on **LLNeRF** with **23.49 dB PSNR**, outperforming the next best method by **+1.58 dB**, along with the highest SSIM (**0.8978**) and lowest LPIPS (**0.3325**). On **LLRS**, we obtain **15.54 dB PSNR (+0.17 dB over I2-NeRF)**, with competitive SSIM and reduced LPIPS. On LOM, while **I2-NeRF** reports higher PSNR due to additional supervision, it leverages ground-truth images to explicitly align luminance and contrast statistics, going beyond point cloud initialization. Despite this advantage, our method still outperforms comparable baselines such as **Aleth-NeRF** and **Lita-GS** by up to **+1.41 dB** and **+0.98 dB**, respectively (Table 1). Overall, our method generalizes consistently across datasets and lighting conditions, demonstrating robust low-light 3D reconstruction without ground-truth supervision.

We further evaluate on the MVTV dataset, which contains high dynamic range 16-bit inputs under extreme low-

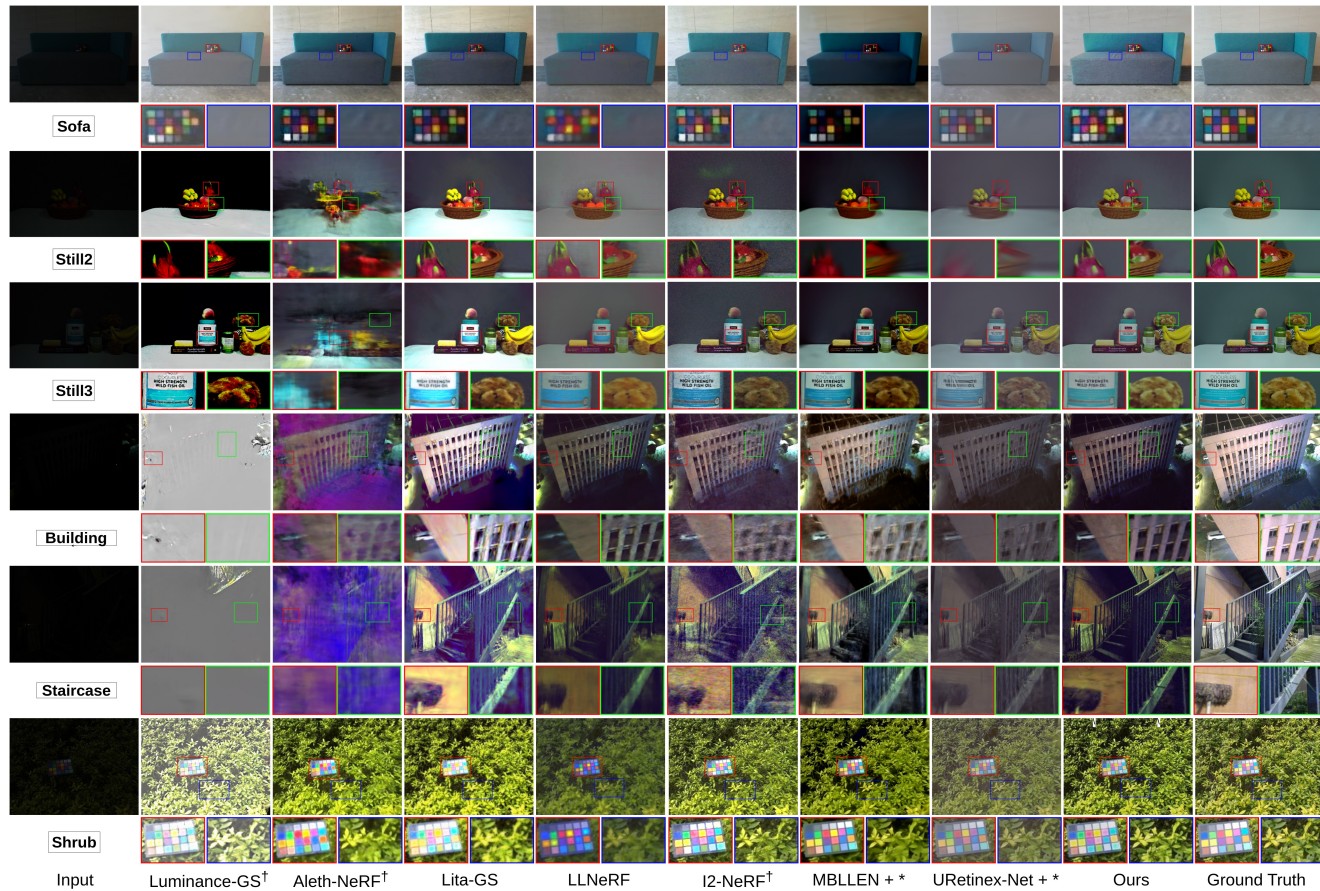

| Input | Luminance-GS$^\dagger$ | Aleth-NeRF$^\dagger$ | Lita-GS | LLNeRF | I2-NeRF$^\dagger$ | MBLLEN + * | URetinex-Net + * | Ours | Ground Truth |

Figure 2. **Qualitative comparison of novel view synthesis under challenging low-light conditions.** Indoor scenes: Sofa (LOM), Still2 and Still3 (LLNeRF); outdoor scenes: Building and Staircase (LLRS), Shrub (LOM). † denotes methods using ground-truth (well-lit) images, * indicates 3DGS-MCMC [11]. Our method achieves competitive visual quality and demonstrates strong generalization across diverse indoor and outdoor datasets under severe noise and extreme low-light conditions.

light conditions (Fig. 3). Inputs are converted to 8-bit via linear tone mapping before applying all 2D and 3D baselines; however, none produce reliable results. Applying gamma correction (GC) and Drago tone mapping followed by 3DGS improves visibility but introduces artifacts. In contrast, our method achieves superior visual quality with better noise suppression and structural consistency. Thus, our methods generalises well across various datasets.

## 5.5. Ablation Study

Our full model achieves significant improvements over all ablated variants, demonstrating the importance of each component. Specifically, it improves PSNR by +2.15 dB over w/o SH regularization, +2.41 dB over w/o geometric loss, and +3.22 dB over w/o N2V + SH regularization. These gains indicate that both radiance modeling and geometric constraints play a crucial role, while the N2V-based denoising is particularly important under extreme low-light conditions. Consistent trends are observed in SSIM, with

improvements of +0.0139, +0.0317, and +0.2044, reflecting better structural fidelity and image consistency. Furthermore, LPIPS decreases by 0.2299, 0.0275, and 0.3424, respectively, showing that our method produces perceptually closer results to ground truth. Notably, the largest degradation occurs when the N2V loss is removed along with SH regularization, highlighting that effective noise suppression is critical for stabilizing training and recovering meaningful details in low-light scenarios. Overall, each component improves reconstruction accuracy and perceptual quality, with ablations confirming the importance of noise-aware regularization and geometric consistency.

## 6. Conclusion

We presented a unified framework for low-light 3D Gaussian Splatting that tightly integrates image enhancement, self-supervised denoising, and geometry-aware optimization. By incorporating noise-aware priors and multi-view consistency directly into the optimization process, our

Table 1. **Quantitative evaluation of novel view synthesis across three low-light datasets.** Our method achieves competitive performance on LOM, where $I^2$-NeRF performs best; however, it relies on ground-truth images for luminance alignment and fails to generalize to LLNeRF and LLRS, while our method generalizes well across all datasets. Green, yellow, and red indicate the best, second-best, and third-best performance, respectively. † denotes methods that rely on ground-truth images for pose or point cloud initialization. GS refers to 3DGS-MCMC [11].

| Method | LOM | | | LLRS | | | LLNeRF | | |
|---|---|---|---|---|---|---|---|---|---|
| | PSNR ↑ | SSIM ↑ | LPIPS ↓ | PSNR ↑ | SSIM ↑ | LPIPS ↓ | PSNR ↑ | SSIM ↑ | LPIPS ↓ |
| **2D Enhancement Methods** | | | | | | | | | |
| SCI++ [17] + GS | 11.86 | 0.6026 | 0.3219 | 9.37 | 0.1224 | 0.7797 | 12.82 | 0.5212 | 0.4549 |
| MBLLEN [16] + GS | 15.08 | 0.7008 | 0.3458 | 15.36 | 0.3958 | 0.6608 | 18.09 | 0.7011 | 0.3656 |
| URetinex-Net[33] + GS | 20.29 | 0.8249 | 0.2991 | 15.10 | 0.4012 | 0.6670 | 20.08 | 0.8679 | 0.3946 |
| Zero-DCE [7] + GS | 13.48 | 0.7179 | 0.2880 | 10.23 | 0.2112 | 0.6950 | 14.19 | 0.6789 | 0.4069 |
| **NeRF-based Methods** | | | | | | | | | |
| Aleth-NeRF† [4] | 19.56 | 0.7822 | 0.3113 | 12.65 | 0.4054 | 0.8985 | 16.02 | 0.7558 | 0.6574 |
| LLNeRF [27] | 17.60 | 0.7497 | 0.3654 | 14.93 | 0.3446 | 0.7292 | 18.82 | 0.8597 | 0.3377 |
| I2NeRF† [14] | 22.40 | 0.7877 | 0.2789 | 15.37 | 0.3763 | 0.6493 | 21.91 | 0.6048 | 0.6463 |
| **3D Gaussian Splatting Methods** | | | | | | | | | |
| LuminanceGS† [5] | 17.98 | 0.7893 | 0.3110 | 9.78 | 0.3334 | 0.7953 | 12.49 | 0.2178 | 0.5291 |
| LITA-GS [36] | 19.99 | 0.7988 | 0.3058 | 14.87 | 0.4381 | 0.7234 | 15.50 | 0.8515 | 0.3973 |
| **Ours** | 20.97 | 0.7904 | 0.2467 | 15.54 | 0.4088 | 0.6136 | 23.49 | 0.8978 | 0.3325 |

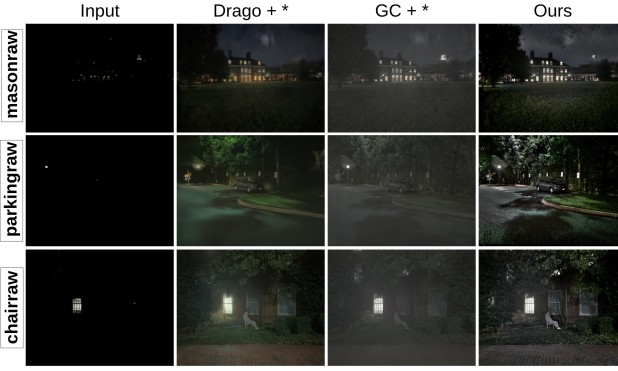

Figure 3. **Qualitative comparison under extreme low-light conditions on the MVTV dataset.** Our method achieves superior visual quality with improved noise suppression and structural consistency under severe low-light conditions. Due to the high dynamic range and extremely low illumination of MVTV, 16-bit inputs are converted to 8-bit via linear tone mapping before applying all 2D and 3D baseline methods. Although gamma correction ($\gamma = 0.2$) and Drago tone mapping followed by 3DGS improve visibility, they still introduce artifacts and inconsistent geometry. Existing baseline methods similarly fail to produce reliable reconstructions under such challenging conditions. (*) indicates the use of 3DGS-MCMC [11].

Table 2. Ablation study on novel view synthesis for the LLNeRF dataset (average over Still2, Still3, and Still4 scenes).

| Method | PSNR ↑ | SSIM ↑ | LPIPS ↓ |
|---|---|---|---|
| w/o SH Reg | 21.34 | 0.8839 | 0.5624 |
| w/o Geometric Loss | 21.08 | 0.8661 | 0.3600 |
| w/o N2V & SH Reg | 20.27 | 0.6934 | 0.6749 |
| **Ours** | **23.49** | **0.8978** | **0.3325** |

evaluations, despite not using ground-truth images for pose estimation or hyperparameter tuning. Future work will explore improved robustness under extreme sensor noise and extend the framework to dynamic scenes and real-time deployment scenarios.

method effectively mitigates noise amplification and improves reconstruction quality under extreme low-light conditions. Extensive experiments demonstrate that our approach generalizes well across diverse datasets, achieving competitive results with existing NeRF-based and Gaussian-based methods in both quantitative and qualitative

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

# NOVA-GS: Noise-Aware View-Consistent Gaussian Splatting for Low-Light Novel View Synthesis

## Supplementary Material

## A. Datasets

Our evaluation is conducted across 39 diverse scenes sourced from four primary datasets: 10 from **MVTV** (16-bit), 16 from **LLNeRF** (where 3 scenes include ground truth), 8 from **LLRS**, and 5 from **LOM**. These datasets cover a spectrum of challenging lighting conditions, from moderate low-light to extreme darkness.

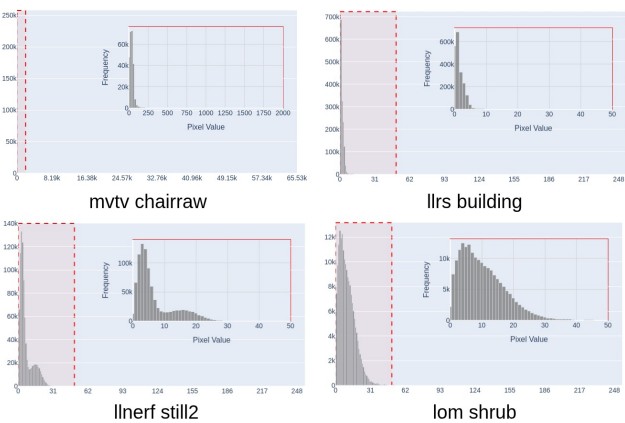

**Figure 4. Pixel intensity distributions across datasets.** MVTV (16-bit) spans a wide dynamic range under extreme low-light conditions; LLRS exhibits extremely low-light characteristics; LL-NeRF represents low-light scenes with noise; and LOM corresponds to moderately low-light scenes. Insets show zoomed low-intensity regions.

The intensity distributions of these datasets, as visualized in Fig. 4, highlight the varying noise profiles and dynamic ranges our model must navigate to achieve consistent reconstruction.

## B. Implementation Details

### B.1. Differentiable Guided Filter Formulation

The Differentiable Guided Filter (DGF) generally computes a filtered output image, denoted as $q$, from an input image $p$, under the structural guidance of a reference image $I$. Let $q_i$, $p_i$, and $I_i$ represent the pixel values of the output, input, and guidance images at spatial index $i$, respectively.

The filter assumes a local linear model between the guidance image and the filtering output. For a local square window $\omega_k$ centered at pixel $k$, the output pixel $q_i$ is modeled as:

$$q_i = a_k I_i + b_k, \quad \forall i \in \omega_k \qquad (18)$$

To determine the linear coefficients $a_k$ and $b_k$, we minimize the reconstruction error between the modeled output and the true input $p_i$, subject to a regularization parameter $\epsilon$:

$$E(a_k, b_k) = \sum_{i \in \omega_k} \left( (a_k I_i + b_k - p_i)^2 + \epsilon a_k^2 \right) \qquad (19)$$

In our self-guided enhancement setup, the guidance image and the input image are identical, taking the value of the dark input view ($I = p = I_{dark}$). Under this condition, the closed-form solutions for the coefficients simplify to:

$$a_k = \frac{\sigma_k^2}{\sigma_k^2 + \epsilon}, \quad b_k = \mu_k(1 - a_k) \qquad (20)$$

where $\mu_k$ and $\sigma_k^2$ are the mean and variance of $I$ within the window $\omega_k$.

Because a single pixel $i$ is covered by multiple overlapping windows, the final pixel value $q_i$ is obtained by averaging the coefficients over all windows containing $i$:

$$q_i = \bar{a}_i I_i + \bar{b}_i \qquad (21)$$

This final output $q$ directly forms the low-frequency brightness map $L$ utilized in our enhancement module. In our implementation, we utilize a window size of $33 \times 33$. Furthermore, rather than utilizing a fixed regularization parameter, we dynamically estimate $\epsilon$ for each image based on the underlying noise variance to adapt to varying degradation levels.

### B.2. Visual Validation of Structure-Aware Enhancement

To visually demonstrate the necessity of our DGF formulation, Figure 5 compares our structure-aware approach against a naive direct pixel-level enhancement. As seen in the bottom row, directly scaling the pixel intensities of the extreme low-light input ($I_{dark}$) amplifies spatially independent sensor noise alongside the signal. This leads to a highly chaotic and fragmented ratio map ($R_{direct}$) and an over-exposed, noisy output with inferior quantitative metrics (PSNR: 18.79 dB, SSIM: 0.7383).

In contrast, our proposed method (top row) derives the enhancement ratio from the DGF's low-frequency brightness output. This smooths out local noise fluctuations while precisely preserving structural boundaries. Applying this stable, structure-aware ratio map multiplicatively prevents uncontrolled noise amplification, yielding a significantly

cleaner enhanced image (PSNR: 19.48 dB, SSIM: 0.7481). This noise-suppressed initialization provides a mathematically stable foundation essential for the downstream 3DGS geometry optimization.

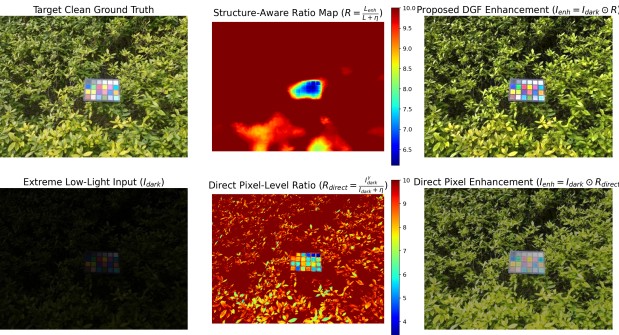

Figure 5. Visual comparison of enhancement strategies. Top row: Our proposed DGF-based enhancement produces a smooth, structure-aware ratio map, mitigating sensor noise amplification and achieving superior image quality. Bottom row: Direct pixel-level enhancement indiscriminately amplifies high-frequency noise, resulting in a chaotic ratio map and degraded quantitative metrics.

## B.3. Deep Attention-ResUNet Architecture Details

The network consists of a four-level encoder (channel depths: 32, 64, 128, 256), a 512-channel bottleneck, and a symmetric decoder.

- **Encoder:** Each level applies Reflection Padding, a $3 \times 3$ Convolution, Instance Normalization, and a LeakyReLU ($\alpha = 0.2$) activation. Downsampling is performed via $2 \times 2$ MaxPool.
- **ResBlocks & Attention:** Each encoder and decoder level contains a ResBlock with two sequential sets of Reflection Padding, $3 \times 3$ Convolutions, and Instance Normalization. A Squeeze-and-Excitation (SE) module (Adaptive Average Pool $1 \times 1$, reduction ratio 16, ReLU, and Sigmoid gating) is applied prior to the residual addition.
- **Decoder & Output:** The decoder uses Bilinear Upsampling (scale factor 2) and $1 \times 1$ Convolutions for channel reduction, concatenated with encoder skip connections.

During training, the Targeted Blind-Spot Masking samples exactly 60,000 pixels per view, replacing them with adjacent pixels from a local $5 \times 5$ neighborhood.

## B.4. Confidence Network Details

The lightweight confidence network consists of two $3 \times 3$ convolutional layers (16 channels) separated by Instance Normalization and a LeakyReLU ($\alpha = 0.2$) activation. The network terminates in a $1 \times 1$ convolution with a Sigmoid activation to scale the final confidence map $C \in [0, 1]$.

## B.5. Results

The qualitative comparisons in Fig. 6 highlight the effectiveness of our method under challenging low-light conditions across multiple datasets. Prior 3D and hybrid 2D+3D approaches either fail to sufficiently suppress noise or introduce artifacts such as over-smoothing and structural distortions. In contrast, our method produces cleaner reconstructions with reduced noise while preserving fine structural details and edges.

Notably, in extremely low-light scenes such as Still4 (LLNeRF) and Stone (LLRS), competing methods struggle with noise amplification and degraded textures, whereas our approach maintains sharper structures and more stable appearance. This demonstrates the advantage of our noise-aware enhancement strategy, which provides a reliable initialization for downstream 3D Gaussian Splatting optimization. Overall, the results indicate strong generalization across diverse noise levels and illumination conditions.

Figure 7 further demonstrates the robustness of our method across a broader set of scenes and object categories. Compared to prior 3D methods, our approach consistently preserves color fidelity, structural integrity, and fine-grained textures, even under severe low-light degradation.

In scenes such as Bike and Buu (LOM) and Pole and Chair (LLRS), existing methods often suffer from color inconsistencies, residual noise, or blurred structures. In contrast, our method achieves more visually coherent reconstructions with improved contrast and reduced artifacts. Additionally, in D5 and White-Chair (LLNeRF), our approach better maintains object boundaries and surface details, highlighting its ability to handle both noise-dominated and signal-sparse conditions.

These results reinforce that the proposed framework effectively balances enhancement and preservation, enabling reliable novel view synthesis across diverse low-light scenarios.

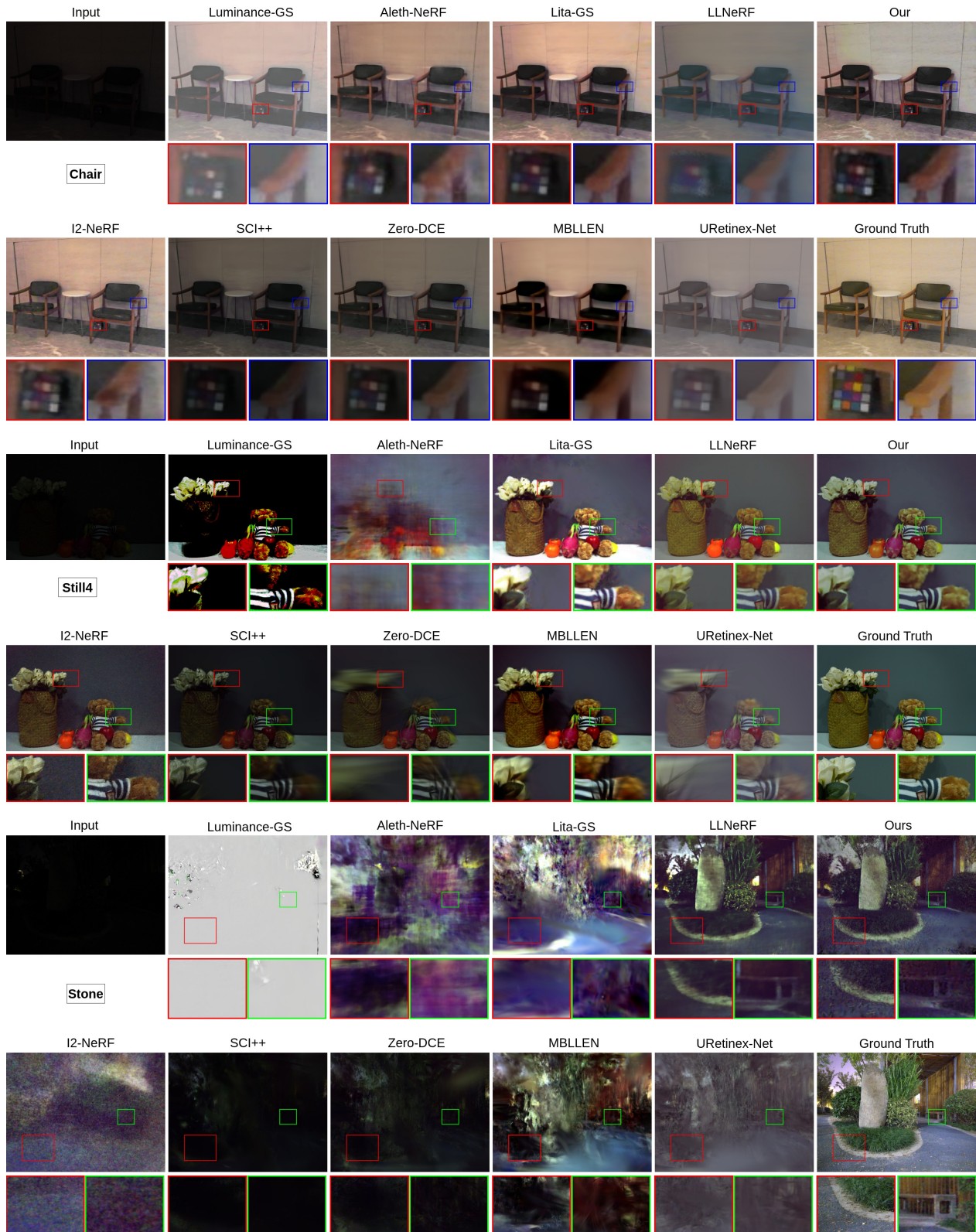

Figure 6. Qualitative comparison of novel view synthesis under challenging low-light conditions. We evaluate on Chair (LOM), Still4 (LLNeRF), and Stone (LLRS). Methods marked with * denote 3DGS-MCMC. Compared to prior 3D and hybrid 2D+3D enhancement methods, our approach produces cleaner reconstructions with reduced noise and improved structural fidelity, demonstrating strong generalization across diverse low-light scenarios.

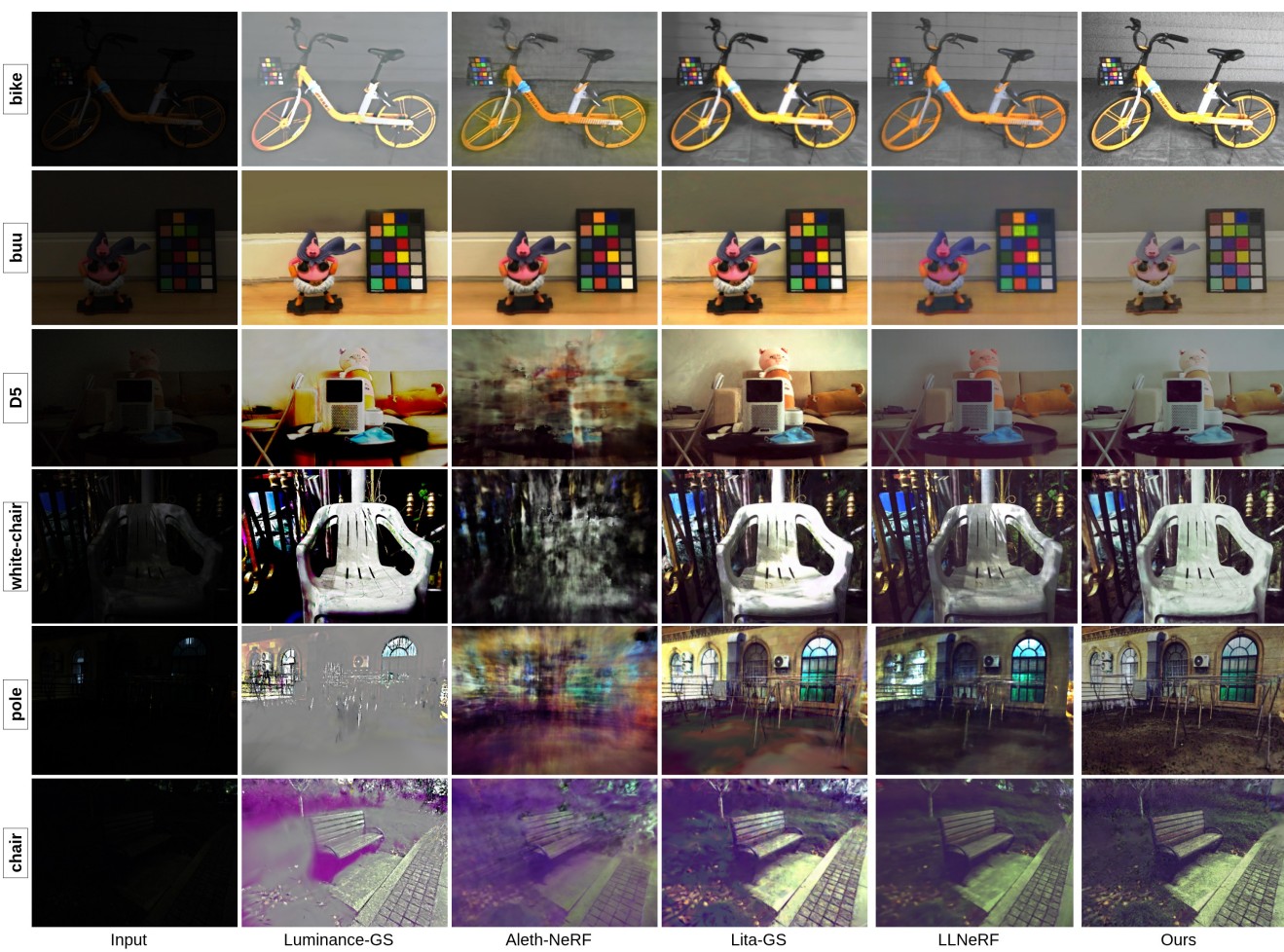

Figure 7. Qualitative comparison of novel view synthesis under challenging low-light conditions. We evaluate on Bike and Ball (LOM), D5 and White-Chair (LLNeRF), and Pole and Chair (LLRS). Compared to prior 3D methods, our approach better preserves color consistency, structural details, and fine textures under severe low-light conditions.