# OpenReview forum: "NOVA-GS: Noise-Aware View-Consistent Gaussian Splatting for Low-Light Novel View Synthesis"
_thecvf.com/CVPR/2026/Workshop/3D4S — CVPR 2026 Workshop 3D4S Oral_

### Official Review · Reviewer_oq9X · 2026-04-14
**Novel modularized GS capture for low light and noisy images**

**Rating:** 8
**Confidence:** 4

**Review:**

The authors propose a modularized approach to handling low-light, noisy images via some novel proposals like Differentiable Guided Filter for noise removal, Targeted Blind-Spot Masking and regularization using Confidence-Aware Photometric Loss and Noise-Guided Spherical Harmonic Regularization. I think using VGGT instead of COLMAP has been adopted widely but the paper introduces other novel aspects which can be used in other pipelines as they are independent modules. The paper/supplementary however misses the ablation experiments to quantize the importance of each of the modules

---

### Official Review · Reviewer_oipu · 2026-04-18
**NOVA-GS: Noise-Aware View-Consistent Gaussian Splatting for Low-Light Novel View Synthesis**

**Rating:** 7
**Confidence:** 4

**Review:**

### Summary
This paper tackles the challenging problem of low-light 3D reconstruction and novel view synthesis. The authors propose a unified framework that combines initialization, enhancement, denoising, and reconstruction into a single pipeline built on 3D Gaussian Splatting. The goal is to improve robustness under severe noise and poor illumination without relying on well-lit reference images or traditional SfM pipelines.

The paper is technically solid and presents a complete system with multiple interacting components. The overall design is well-motivated, and the integration of denoising and reconstruction is handled carefully. The experiments are reasonably thorough, covering several datasets and including ablation studies. That said, the performance gains over prior methods are relatively modest, and the evaluation could be strengthened with more rigorous comparisons and deeper analysis of robustness.

The paper is generally well-organized, and the high-level pipeline is easy to understand. However, some sections are overly verbose and could be streamlined. A few components are not explained in sufficient detail, which makes it harder to fully assess their contribution. Improving the writing and reducing repetition would significantly enhance readability.

The work mainly builds on existing ideas and combines them into a unified framework. While the integration is thoughtful, most individual components (e.g., denoising strategy, enhancement, geometric consistency) are not fundamentally new. The noise-aware regularization is a nice touch and adds value, but overall the contribution feels more like a strong engineering effort rather than a conceptual breakthrough.

The problem is important and relevant, especially for applications in robotics and real-world scene understanding under challenging lighting conditions. Removing the dependence on SfM and clean supervision is a meaningful step forward. However, the impact is somewhat limited by the incremental nature of the contributions and the lack of deeper insights.

### Pros
- Addresses a relevant and challenging real-world problem
- Presents a unified and coherent framework
- Removes dependence on SfM and well-lit supervision
- Demonstrates consistent improvements across datasets
- Includes useful ablation studies

### Cons
- Limited conceptual novelty
- Improvements over prior work are moderate
- Some important baselines are missing
- Certain components are under-explained
- Writing could be clearer and more concise
- Limited discussion of failure cases

Overall, this is a solid and well-executed paper with clear practical value. While it does not introduce fundamentally new ideas, the integration of existing techniques is done carefully and leads to consistent improvements. With stronger evaluation and clearer presentation, the paper could be more compelling.

---

### Decision · Program_Chairs · 2026-04-28

Accept (Oral)